# A Comparison of Skin Lesions’ Diagnoses Between AI-Based Image Classification, an Expert Dermatologist, and a Non-Expert

**DOI:** 10.3390/diagnostics15091115

**Published:** 2025-04-28

**Authors:** Lior Mevorach, Alessio Farcomeni, Giovanni Pellacani, Carmen Cantisani

**Affiliations:** 1Dermatology Unit, Department of Clinical Internal Anesthesiological and Cardiovascular Sciences, “Sapienza” University of Rome, 00161 Rome, Italy; dermatologysapienza@gmail.com (L.M.); pellacani.giovanni@uniroma1.it (G.P.); 2Faculty of Economics, Tor Vergata University of Rome, 00133 Roma, Italy

**Keywords:** AI-based classification, dermatology, skin lesions, diagnostic accuracy, image recognition, machine learning

## Abstract

**Background/Objectives**: This study aims to evaluate and compare the diagnostic accuracy of skin lesion classification among three different classifiers: AI-based image classification, an expert dermatologist, and a non-expert. Given the rising interest in artificial intelligence (AI) within dermatology, it is crucial to assess its performance against human expertise to determine its viability as a reliable diagnostic tool. **Methods**: This reader study utilized a set of pre-labeled skin lesion images, which were assessed by an AI-based image classification system, an expert dermatologist, and a non-expert. The accuracy of each classifier was measured and compared against the ground truth labels. Statistical analysis was conducted to compare the diagnostic accuracy of the three classifiers. **Results**: The AI-based image classification system exhibited high sensitivity (93.59%) and specificity (70.42%) in identifying malignant lesions. The AI model demonstrated similar sensitivity and notably higher specificity compared to the expert dermatologist and non-expert. However, both the expert and non-expert provided valuable diagnostic insights, especially in classifying specific cases like melanoma. The results indicate that AI has the potential to assist dermatologists by providing a second opinion and enhancing diagnostic accuracy. **Conclusions**: This study concludes that AI-based image classification systems may serve as a valuable tool in dermatological diagnostics, potentially augmenting the capabilities of dermatologists. However, it is not yet a replacement for expert clinical judgment. Continued improvements and validation in diverse clinical settings are necessary before widespread implementation.

## 1. Introduction

### 1.1. Epidemiology

Skin lesions encompass a wide range of types, categorized as pigmented or non-pigmented and further classified as melanocytic or non-melanocytic based on their cellular origin. Melanocytic lesions arise from melanocytes, the pigment-producing cells found in the epidermis, dermoepidermal junction, and hair follicles. Non-melanocytic lesions originate from keratinocytes, fibroblasts, or blood vessels, sometimes exhibiting increased melanin production [1]. Both types can manifest as benign or malignant tumors.

Among malignant skin tumors, Basal Cell Carcinoma (BCC) and Squamous Cell Carcinoma (SCC), both of non-melanocytic origin, are the most common. Despite their prevalence, mortality rates remain relatively low, with annual deaths in the United States estimated between 2000 and 8000, mostly from SCC [2]. In contrast, Melanoma, a malignant melanocytic lesion, accounts for only 1% of diagnosed skin cancers but has a significantly higher fatality rate due to its potential for rapid metastasis. Global melanoma-related deaths are estimated at 60,000 annually [3], with projections of 8430 deaths in the US in 2025 alone, primarily among men [4]. Advances in early diagnosis and treatment have contributed to declining melanoma death rates in recent years [5], but early detection remains critical to preventing disease progression.

Differentiating melanoma from other pigmented lesions is challenging, as approximately 33% of melanomas develop from preexisting benign nevi [6]. This highlights the importance of accurate diagnosis and careful monitoring of pigmented lesions. Non-invasive diagnostic tools, including dermoscopy, reflectance confocal microscopy (RCM), and optical coherence tomography (OCT) [7], play a vital role in improving diagnostic accuracy. Dermoscopy, in particular, allows for magnified visualization of skin surface structures, aiding in the differentiation of various tumor types and enhancing clinical decision-making.

### 1.2. Types of Skin Tumors

#### 1.2.1. Pigmented Lesions

##### Pigmented Benign Melanocytic Lesions

Benign melanocytic lesions are non-cancerous proliferations of melanocytes but can resemble melanoma, requiring careful evaluation. They often exhibit symmetry, slow growth, and lack spontaneous bleeding [8].

Nevi (Moles): Common benign lesions appearing as well-demarcated pigmented macules or papules. Dermoscopic features include globular, reticular, cobblestone, or homogeneous patterns [9].Lentigines: Flat, uniformly pigmented brown macules on sun-exposed areas. While generally benign, they may rarely progress to melanoma [10,11].Dysplastic Nevi: Atypical nevi with irregular borders, variegated coloration, and asymmetry. Though mostly benign, they indicate an increased melanoma risk [12].

##### Pigmented Benign Non-Melanocytic Lesions

Pigmented Benign Keratoses: Includes seborrheic keratosis and lichenoid keratosis, which mimic melanocytic lesions. Dermoscopic features like comedo-like openings and milia-like cysts help differentiate them [13,14].Dermatofibroma: Firm, raised nodules with pink to brown coloration. Can mimic melanoma due to irregular borders and variable pigmentation [15,16].

##### Pigmented Malignant Melanocytic Lesions

Melanoma: Classified into in situ and invasive types, with subtypes including superficial spreading, nodular, lentigo maligna, acral lentiginous, and desmoplastic melanoma. Common dermoscopic features include asymmetry, irregular streaks, and multiple colors.

##### Pigmented Malignant Non-Melanocytic Lesions

Basal Cell Carcinoma (BCC): A slow-growing tumor often found on sun-exposed areas. The pigmented variant lacks a pigment network but exhibits telangiectasia, blue-gray ovoid nests, and spoke wheel areas [17,18].Squamous Cell Carcinoma (SCC): Arises from keratinocytes, often from actinic keratosis. Presents as irregular scaly plaques with glomerular vessels, brown globules, and homogeneous pigmentation [19,20].

#### 1.2.2. Non-Pigmented/Amelanotic Lesions

##### Non-Pigmented Benign Lesions

Non-pigmented benign lesions can resemble amelanotic melanoma, complicating diagnosis.

Dermal Nevi, Dermatofibroma, Seborrheic Keratosis: Can appear flesh-colored, pink, or tan, requiring careful dermoscopic assessment [21,22,23,24].Actinic Keratosis (AK): Precancerous lesions from sun exposure. Facial AK shows erythematous vessel pseudo network and follicular openings with white halos [25].

##### Non-Pigmented Malignant Melanocytic Lesions

Amelanotic Melanoma: Lacks pigmentation, presenting as an asymmetrical pink or red lesion. Dermoscopic features include irregular vessel clusters, reticular depigmentation, and scattered pigmented remnants [26]. Early detection is critical.

##### Non-Pigmented Malignant Non-Melanocytic Lesions

BCC: Appears as translucent papules with telangiectasia, sometimes ulcerated or crusted. Crystalline structures and white shiny lines are common under polarized dermoscopy [17].SCC: Presents as scaly pink lesions with glomerular vessels, white perifollicular circles, and variable vasculature. Ulceration and irregular morphology make identification challenging [20].

### 1.3. Dermoscopy

Advancements in dermatological diagnostic tools have significantly improved clinicians’ ability to identify and characterize skin lesions. While naked-eye examination remains essential, dermoscopy has revolutionized the field by enhancing visualization and diagnostic accuracy.

A dermoscope, also known as a dermatoscope, is a handheld device used for non-invasive skin examination. It consists of a magnifying lens and a built-in light source, often featuring polarized and non-polarized settings, adjustable magnification, and digital imaging for documentation. By providing a magnified view, dermoscopy aids in identifying pigmented and non-pigmented lesions, including melanoma. Key dermoscopic features assessed include:Pigment network—Arrangement of pigmented lines or networks within the lesion.Structureless areas—Lack of discernible structures or patterns.Vascular patterns—Presence of blood vessels such as linear, dotted, or serpentine vessels.Symmetry—Evaluation of structural symmetry or asymmetry.Colored structures—Presence of specific colors like white lines, dots, or varying shades of black, brown, blue, or white.

Dermoscopy dates back to the Middle Ages (c. 1650) when Pierre Borel studied nail bed capillaries under a microscope. Later, Carl Hueter and Ernst Karl Abbe advanced immersion microscopy with cedar oil, improving resolution. In 1893, Unna emphasized the importance of skin translucency in surface microscopy. The early 20th century saw further advancements by Muller, who developed capillary microscopes with magnifications of 10–172×.

Saphier coined the term “dermatoscopy”, initially studying skin capillaries and pigmentation. Jeffrey C. Michael introduced dermoscopy to the U.S. in 1922, and in the 1950s, Lean Goldman applied it to melanocytic nevi and melanoma despite technological limitations. A key milestone came in 1971 when Dr. Ronald Mackie advocated its use for preoperative diagnosis of pigmented lesions, particularly melanoma. Throughout the 1980s and 1990s, Kreusch and Rassner developed the first handheld dermoscope, spurring further dermatological advancements.

The late 20th century brought digital technology, enabling videodermoscopy and teledermoscopy, significantly expanding dermoscopy’s accessibility [27]. In 2011, the Consensus Net Meeting at the First World Congress of Dermoscopy in Rome introduced standardized methodologies for diagnosing pigmented lesions:Pattern analysis—Systematic evaluation of dermoscopic structures and patterns.ABCD rule—Asymmetry, border irregularity, color variation, and dermoscopic structures for melanoma assessment.Menzies method—Systematic assessment of features like color variation, symmetry, and irregular structures.Seven-point checklist—Evaluation of atypical vascular patterns, irregular streaks, blotches, dots/globules, and regression structures [28].

Dermoscopy has significantly improved the diagnosis and management of skin lesions, enhancing early melanoma detection and patient outcomes. It serves as a preliminary tool to reduce invasive procedures, alleviate patient stress, and lower healthcare costs while ensuring timely treatment. Complementary assessments, such as Reflectance Confocal Microscopy (RCM) and histopathological analysis, further refine the diagnosis. As a cost-effective, efficient, and user-friendly technique, dermoscopy continues to evolve, bolstering dermatological diagnostics through continuous innovation.

### 1.4. Diagnostic Process of Skin Lesions

The diagnostic process of skin lesions, particularly melanoma, encompasses a comprehensive clinical examination that should be started with a total body visual skin inspection, using both naked-eye examination and dermoscopy, to detect both pigmented and non-pigmented lesions. Dermoscopy should be applied to all lesions, not only to clinically suspicious ones, in order to unveil features suggestive of malignancy that may be clinically invisible to the naked eye. If a pigmented lesion is raised or exhibits multiple dermoscopic features suggestive of melanoma, immediate biopsy or reflectance confocal microscopy (RCM) is recommended.

#### 1.4.1. Dermoscopic Features

Dermoscopy aids in identifying melanoma through features such as an atypical network, focal streaks, irregular dots, a blue-white veil, and vascular structures. Specific melanoma subtypes exhibit distinct dermoscopic patterns:Superficial Spreading Melanoma (SSM): Chaotic structure, atypical network, irregular hyperpigmentation.Nodular Melanoma (NM): Predominantly nodular, with blue pigmentation and vascular structures.Lentigo Maligna Melanoma (LMM): Gray rhomboids, non-evident follicles, intense pigmentation.Acral Lentiginous Melanoma (ALM): Parallel ridge, furrow, or fibrillar pattern.Desmoplastic and Amelanotic Melanoma: Scar-like appearance, polymorphic vascular pattern.Melanoma of the Nail Matrix: Irregular pigmentation lines, Hutchinson signs.Mucosal Melanoma: Multicolored appearance (brown, black, blue, red, white, gray) [29,30].

#### 1.4.2. Melanoma Risk Stratification Algorithms

Several algorithms have been developed to aid both expert dermatologists and non-expert clinicians in identifying potential melanomas and guiding further assessment to decide whether an excision is required. The following melanoma risk stratification algorithms are utilized in clinical practice to aid macroscopic and dermoscopic evaluation in the diagnostic process.

ABCDE Rule—a fundamental tool for assessing pigmented lesions, with each letter representing a distinct criterion. It was found to be more commonly indicative of SSM.

Asymmetry: Evaluate whether the lesion exhibits symmetrical or asymmetrical characteristics in shape and structure. Asymmetry is determined along perpendicular axes, with a score of 1 assigned if there is asymmetry observed in either the outer shape or the differential structures along at least one axis. The maximum score attainable is 2, indicating pronounced asymmetry.Border irregularity: Examine the edges of the lesion for irregular, poorly defined, or jagged borders. Lesions are divided into four circumferential segments for border score calculation. A score of 1 is assigned if there is an abrupt cutoff of the network observed in at least one-quarter of the lesion’s circumference.Color variation: Determine the color score by assessing the number of distinct colors present within the lesion. The colors considered include white, red, light and dark brown, blue-gray, and black. A score of 1 is assigned if there are at least three different colors present within the lesion.Diameter: Measure the size of the lesion to determine if it exceeds 6 mm, potentially indicating an increased risk of melanoma. Note that melanomas can manifest at smaller sizes as well [31].Evolution: Evaluate any changes in the lesion’s size (horizontal enlargement), shape, color, surface characteristics (e.g., bleeding), or accompanying symptoms (e.g., itching, tenderness) reported by the patient over the past three months. Scores range from −1, indicating no change, to +1, indicating observable changes during this period.

A total ABCD score of at least 4 suggests a higher likelihood of melanoma [32,33].

EFG Rule—focuses on evaluating nodular lesions, which are more frequently amelanotic/hypomelanotic. Each letter represents a characteristic that together should raise suspicion for melanoma:6.Elevated.7.Firm on palpation.8.Growing progressively for more than a month [34].

Ugly Duckling Sign—this sign involves comparing lesions within an individual to identify any deviations from the usual nevus characteristics, both clinically and dermoscopically. Lesions that stand out or differ from the individual’s typical nevi pattern warrant special attention, regardless of meeting traditional ABCDE criteria [35,36].

Little Red Riding-Hood Sign—refers to a lesion that clinically resembles other nevi within an individual but, in dermoscopic examination, has highly suspicious dermoscopy features. It may have characteristics that are commonly observed in amelanotic melanomas, such as pink, reddish, or purplish–red coloration that are often associated with vascular patterns [37,38].

Menzies Method—this method incorporates the following characteristics:More than one distinct color present within the lesion.Asymmetrical pattern.At least one positive feature of the following: blue–white veil, multiple brown dots, pseudopods, radial streaming, scar-like depigmentation, peripheral black dots/globules, multiple colors (5 or 6), multiple blue/gray dots, broad pigment network [39].

7-Point Checklist—this checklist comprises major and minor dermoscopic criteria associated with melanoma, with a total score of three or above suggesting a higher likelihood of melanoma.

Major criteria (2 points): atypical pigment network, gray-blue areas, atypical vascular pattern.Minor criteria (1 point): Radial streaks, irregular diffused pigmentation (blotches), irregular dots/globules, regression pattern (white (scar-like) areas, hypopigmented areas, peppering (multiple gray-blue dots)) [40].

Glasgow 7-Point Checklist—this checklist incorporates clinical signs, dermoscopic features, and symptoms frequently associated with melanoma, with a total score of three or above suggesting a higher likelihood of melanoma.

Major criteria (2 points): change in size, irregular pigmentation, irregular borders.Minor criteria (1 point): inflammation, itch/altered sensation, larger than other lesions (diameter > 7 mm), oozing/crusting of the lesion [41] (p. 7, [42]).

3-Point Checklist—a checklist designed for non-experts, evaluates asymmetry, atypical network, and blue-white structures to identify high-risk melanomas [43].

Chaos and Clues Algorithm—this algorithm involves evaluating lesions for “chaos” (asymmetry of structure or color) and identifying eight clues to malignancy: eccentric structureless zones of any color (except skin color), gray or blue structures, black dots or clods in the periphery, segmental radial lines or pseudopods at the periphery, white lines, thick reticular lines, polymorphous vessels, and parallel lines on the ridges (for acral lesions). If no chaos is present, the examiner moves to the next lesion [30].

These algorithms are crucial for early detection, particularly in underserved areas where dermatologists are scarce, enabling timely referrals and optimizing healthcare resources.

#### 1.4.3. Peri- and Post-Dermoscopic Assessments

The following assessment procedures are recommended for high-risk adults or concerning lesions, serving as confirmatory steps after the initial dermoscopic assessment. High-risk factors for melanoma include conditions like Familial Atypical Multiple Mole Melanoma syndrome (FAMMM), Dysplastic Nevus syndrome, large congenital nevi, elderly patients with sun-damaged skin, and those with a personal or family history of melanoma.

Total-Body Photography (TBP) involves high-resolution photos of the entire body, documenting suspicious lesions. These images serve as a baseline for future assessments, tracking lesion evolution over time. Automated 3D TBP offers more standardized, high-quality images compared to other systems that may vary by operator or device [44].

Sequential Digital Dermoscopic Imaging (SDDI) is used to monitor atypical lesions. Recommended for clinically flat lesions or those with minimal dermoscopic features of melanoma, SDDI stores images for serial observation. Lesions under surveillance should be re-evaluated after three months; any significant changes warrant biopsy [29].

Reflectance Confocal Microscopy (RCM) is a non-invasive imaging technique for visualizing lesions at the cellular level, particularly useful for challenging cases like amelanotic melanoma. RCM is indicated for equivocal lesions on sensitive areas, or for patients reluctant to undergo biopsy [45].

Pigmented Lesion Assay (PLA) is a molecular test that assesses melanoma-specific genes from a sample of the stratum corneum using an adhesive patch. Positive results indicate the need for a biopsy. PLA is suitable for equivocal lesions in sensitive areas or in patients who are hesitant to undergo a biopsy, but it is unnecessary for lesions with well-defined malignant features.

Teledermatology and teledermoscopy allow remote evaluation of concerning pigmented lesions, especially for patients with limited access to dermatologic care. Dermoscopic smartphone attachments enable image capture for consultation, facilitating the monitoring of atypical nevi [29].

Integrating these tools into routine practice enhances early melanoma detection and reduces unnecessary excisions. Suspicious lesions should still be excised for histopathological and molecular analysis to guide treatment.

#### 1.4.4. Difficulties in the Diagnostic Process

The diagnostic process for skin lesions, particularly melanoma, presents several challenges that can affect accuracy and patient outcomes. These difficulties stem from limitations in diagnostic tools, human perception, and interpretation.

A semantic gap arises when there is a discrepancy between dermatological terminology and its interpretation, often due to inexperience or a lack of clinical exposure, which can lead to diagnostic errors, especially among non-expert clinicians. This gap is further exacerbated by the evolving dermoscopic terminologies in the literature [46]. Ongoing training is crucial to improving interpretation accuracy.

Risk assessment tools aim to improve diagnostic accuracy by stratifying lesions based on malignancy likelihood. However, these tools are subjective and may overlook early melanoma signs or misclassify benign lesions as high-risk. Moreover, they often fail to account for individual lesion variations or patient characteristics, leading to diagnostic errors [47].

Diagnostic tools such as dermoscopy are essential but have limitations. Variations in device quality, magnification, and lighting can impact lesion visualization, while certain lesion features may be obscured under pressure [48]. Difficult-to-visualize areas, like the scalp or mucosal surfaces, and artifacts like dirt or hair, can also hinder accurate assessment. Misuse of tools by untrained professionals can result in suboptimal outcomes, and variability in imaging methods can further affect diagnostic accuracy [49].

The semantic gap is especially evident when dermoscopy is used by non-experts. Limited experience in real-time diagnosis leads to confusion and misinterpretation. Dermoscopic diagnosis requires extensive practice and training to develop proficiency.

Even experts can be influenced by biases that affect diagnostic outcomes, such as false positives or negatives, depending on their expertise and the clinical context. External pressures like time constraints can also lead to errors.

Geographical and cultural factors contribute to diagnostic disparities, as clinicians trained predominantly in certain demographic settings may struggle with diagnosing lesions in populations with diverse skin types or ethnic backgrounds. This highlights the need for diversity and inclusion in medical training to ensure equitable patient care [50].

In conclusion, the diagnostic process for skin lesions faces challenges due to semantic gaps, tool limitations, and clinician biases. Addressing these issues requires ongoing education and training to improve diagnostic accuracy and patient outcomes.

### 1.5. AI in Dermatology

The integration of artificial intelligence (AI) and machine learning (ML) in dermatology has revolutionized skin lesion classification, especially melanoma. ML algorithms learn from data through supervised, semi-supervised, or unsupervised approaches. While unsupervised learning identifies patterns without labeled samples, supervised learning uses labeled data for model training, and semi-supervised learning combines both for improved performance.

Various algorithms, including Random Forest, K-Nearest Neighbors (KNN), Support Vector Machine (SVM), and Convolutional Neural Networks (CNN), have been applied for image classification. Rule-based algorithms like Decision Trees (DT) and Naïve Bayes Classifier (NBC) use predefined rules to analyze dermoscopic images, identifying features like asymmetry and border irregularity. However, they may lack the ability to assess local relationships between lesion features, crucial for accurate diagnosis [51].

Among artificial neural networks (ANNs), convolutional neural networks (CNNs) remain the most widely used and effective for dermatological imaging tasks. CNNs automatically learn hierarchical visual features from raw pixel data, enabling precise lesion segmentation and classification without relying on hand-crafted rules. Multiple studies have shown that CNNs can match, and in some cases surpass, dermatologist-level accuracy in distinguishing between benign and malignant lesions [52,53]. However, CNNs require training on large, diverse datasets to generalize reliably, highlighting the need for global data-sharing initiatives and standardized imaging protocols [54].

Transformer-based architectures, such as the Vision Transformer (ViT), have recently been adapted from natural language processing to computer vision tasks. Unlike CNNs, which rely on localized filters, ViTs divide images into patches and apply self-attention to model global contextual relationships. These models have shown comparable or superior performance to CNNs in skin lesion classification, particularly when fine-tuned on dermatology-specific datasets. Their ability to capture long-range dependencies and manage image variability makes them a promising tool for future dermatological AI research [55].

In parallel, the integration of AI with Total Body Photography (TBP) systems has enabled more comprehensive lesion monitoring. The latest 3D TBP CNN-based platforms can generate full-body reconstructions and automatically detect changes in nevi, improving melanoma surveillance. However, limitations persist, including inconsistent performance in automated nevus counting and challenges in real-world deployment [55,56].

Mobile applications with AI algorithms also allow users to capture skin lesion images with smartphones for preliminary analysis. Despite their potential, these apps face challenges in clinical testing and regulatory approval, with only two apps currently approved in Europe [57].

Widespread AI adoption in dermatology faces several challenges, including the need for standardized databases, consistent imaging protocols, and diverse datasets. Variations in imaging equipment, lighting, and techniques can introduce biases, and factors like noise or sun-damaged skin may interfere with AI pattern recognition. Additionally, the underrepresentation of rare melanoma subtypes or darker skin types in training datasets limits AI performance. AI algorithms typically analyze single lesions, missing contextual comparisons like the Ugly Duckling or Little Red Riding Hood signs used in clinical practice. Addressing these issues requires refining AI algorithms and curating representative datasets [54].

AI and ML have shown strong potential in skin lesion classification, especially melanoma. From CNNs to transformer-based models, recent approaches have achieved high diagnostic accuracy. Tools like 3D TBP and mobile apps are expanding clinical use, though challenges remain—such as inconsistent training data, underrepresented skin types, and limited validation across user expertise levels—highlighting the need for continued, targeted evaluation efforts.

### 1.6. The Study

#### Aim of the Study

This study aims to evaluate and compare the diagnostic accuracy of skin lesion classification across three classifiers: an AI-based image classification system, an expert dermatologist, and a non-expert. The primary contributions of this study include:The independent evaluation of DEXI, a proprietary AI model, on a previously untested dataset from an Italian population, providing new insights into its performance in a different demographic.A comprehensive three-way comparison of diagnostic accuracy between DEXI, an expert dermatologist, and a non-expert, offering a unique perspective on AI’s potential in dermatology.The use of real-world dermoscopic images from a high-risk clinical setting to assess diagnostic accuracy, which reflects practical challenges in clinical decision-making.

## 2. Materials and Methods

This study evaluated the diagnostic performance of three classifiers in categorizing skin lesions and distinguishing between benign and malignant: (1) a deep-learning model named Dermoscopy EXplainable Intelligence (DEXI), (2) an expert dermatologist, and (3) a non-expert medical student.

DEXI is an AI-based diagnostic tool developed by Canfield Scientific. It is built on the EfficientNetV2 convolutional neural network (CNN) architecture and is designed to generate a malignancy risk score ranging from 0 to 10 for eight lesion categories, including melanoma. The version used in this study was DEXI 2.1, the latest available at the time of publication. The model is available for research purposes and is not yet approved for clinical use [56,58].

DEXI was trained using a combination of proprietary datasets from Canfield Scientific and publicly available dermoscopic images from the International Skin Imaging Collaboration (ISIC) archive, which includes 81,155 images labeled with both diagnostic classifications and benign/malignant status (Table 1) [59].

The expert classifier was a board-certified dermatologist with 25 years of experience in dermoscopy and clinical dermatology. The non-expert was a medical student who had undergone targeted training for 3 months, including practical exposure to dermoscopic image analysis, review of dermoscopy atlases (i.e., Handbook of Dermoscopy), and guided exploration of the ISIC image archive.

The dataset used for this study consisted of 1047 dermoscopic images corresponding to 302 unique skin lesions. These were randomly selected from a larger dataset of 15,178 images collected at the Dermatology Department of the University of Modena and Reggio Emilia. This dataset offers several distinct advantages compared to other open-source datasets commonly used in dermatological AI research:All lesions have histopathological confirmation, ensuring high diagnostic accuracy.Lesions were selected based on clinical suspicion of malignancy, reflecting real-world diagnostic conditions, and providing a more practical clinical context rather than artificial case selection.The dataset represents a distinct geographic population—Italian patients—which enhances demographic diversity beyond that of widely used datasets such as the ISIC archive, which primarily includes patients from the U.S., Australia, and Spain.The dataset includes multiple images per lesion, ranging from 1 to 23, which allows for a richer and more comprehensive assessment of classifier performance by capturing different visual aspects of each lesion.

The lesions in the dataset include a variety of types: 180 nevi, 54 melanomas, 21 basal cell carcinomas, 3 squamous cell carcinomas, 3 actinic keratoses, 26 benign keratosis-like lesions, 4 dermatofibromas, and 11 of unknown type. All lesions were excised based on clinical, dermoscopic, and/or confocal microscopy findings, suggesting the presence of melanoma or other malignancies. Randomization of images was performed using random.org (accessed on 14 December 2021). While the full dataset cannot be publicly released due to ongoing research constraints, it is available for academic collaboration under restricted access.

Each image in the testing stage was independently evaluated by both the expert and non-expert to determine the lesion diagnosis and classify it as benign or malignant, based solely on dermoscopic visual characteristics. The evaluations were performed blind to the final histopathological diagnosis.

Common dermoscopic criteria were used to guide decision-making. Benign lesions were typically characterized by symmetry, stable color distribution, and well-defined borders. Malignant lesions often showed asymmetry, irregular borders, and multicolor pigmentation (e.g., black, brown, blue, white).

Dermoscopic features of melanoma are characterized by atypical networks, irregular dots or globules, and a blue-white veil, further indicating malignancy. Pigmented BCC is distinguished by patterns such as arborizing vessels, large blue-gray nests, and leaf-like areas, while SCC often shows glomerular vessels and scaly surfaces. Actinic keratosis commonly presents with a pink or red background and white-to-yellowish scales, benign keratosis-like lesions feature milia-like cysts and comedo-like openings, and dermatofibromas show a central white scar-like area with a peripheral pigment ring.

Lesion categories used for classification were standardized as follows: Nevus (NV), Melanoma (MEL), Basal Cell Carcinoma (BCC), Squamous Cell Carcinoma (SCC), Actinic Keratosis (AK), Benign Keratosis-like Lesions (BKL), Dermatofibroma (DF), and Unknown.

Ground truth classification into “benign” or “malignant” was based on histopathological diagnosis and the most common clinical presentation of each lesion type (Table 2). Some lesions (e.g., AK and nevi) presented classification ambiguity due to their precancerous or borderline nature.

Actinic Keratosis (AK) is considered benign in this study, though it is clinically recognized as a precursor to SCC.Nevi (NV) are generally benign, though a small subset may progress to melanoma.

To maintain consistency across all classifiers, ambiguous lesion types were assigned a predetermined classification (e.g., AK as benign). This standardization enabled uniform statistical analysis and accurate performance comparisons.

The DEXI model outputs a malignancy risk score on a scale of 0–10. In this study, a threshold of ≥3.5 was used to classify a lesion as malignant, which was considered a “very high” cut-off. This score represents ten times the combined probability of the three malignant categories (MEL, BCC, SCC). For each lesion, if multiple images were available, DEXI scores and classification probabilities were averaged across all images.

Final classifications by DEXI, the expert, and the non-expert were compared against the ground truth. Diagnostic performance was assessed in terms of binary classification (benign vs. malignant), and further analysis was conducted by lesion type to evaluate accuracy across categories.

## 3. Results

### 3.1. Classification Accuracy

The classification accuracy of each evaluator—DEXI, an expert dermatologist, and a non-expert medical student—was assessed based on their ability to correctly identify lesions as benign or malignant. As shown in Table 3, DEXI achieved the highest overall accuracy, particularly in identifying benign lesions, with 150 correct classifications (70.42%) and 73 malignant lesions correctly identified (93.59%).

Both human classifiers demonstrated strong performance in classifying malignant lesions, with the expert achieving an accuracy of 93.59% (73 correct classifications) and the non-expert slightly higher at 94.87% (74 correct). However, their accuracy in identifying benign lesions was substantially lower: 70 benign lesions (32.86%) for the expert and 61 (28.64%) for the non-expert.

Overall, the combined accuracy in identifying malignant lesions across all classifiers was 94%, which was notably higher than the 44% overall accuracy for benign lesions. The overall accuracy in categorizing lesions as malignant (94%) was notably higher than for benign lesions (44%).

### 3.2. Sensitivity and Specificity

To further evaluate classifier performance, sensitivity and specificity metrics were calculated based on the confusion matrix values (Table 4), with the results summarized in Table 5.

DEXI demonstrated a sensitivity of 0.936, equal to the expert, and only slightly lower than the non-expert’s sensitivity of 0.948, reflecting a high capacity to correctly identify malignant lesions across all classifiers. However, DEXI outperformed both human evaluators in specificity—its ability to correctly identify benign lesions—with a value of 0.704, compared to 0.328 for the expert and 0.286 for the non-expert.

These findings suggest that while all classifiers are similarly effective in detecting malignancy, DEXI exhibits a capability to minimize false positives by more accurately distinguishing benign lesions.

### 3.3. Agreement and ROC Analysis

In addition to sensitivity and specificity, the diagnostic performance of the AI model (DEXI) was further evaluated using the area under the ROC curve (AUC) (Figure 1) and inter-rater agreement metrics between classifiers using Cohen’s kappa statistics (Table 6). DEXI achieved a high AUC of 0.922 (95% CI: 0.881–0.958), confirming its strong overall discriminative ability in distinguishing malignant from benign lesions. Kappa analysis revealed the highest agreement between the expert and non-expert human raters (κ = 0.43, 95% CI: 0.31–0.55), indicating moderate consistency. Agreement between AI and the expert (κ = 0.23, 95% CI: 0.14–0.32) and between AI and the non-expert (κ = 0.28, 95% CI: 0.19–0.36) was lower, suggesting that while AI performance aligns well with ground truth labels, its diagnostic strategy differs from that of human evaluators. These findings highlight the complementary nature of human expertise and AI-driven classification.

### 3.4. Diagnostic Accuracy by Lesion Type

The accuracy of each classifier was also analyzed by specific lesion type, as shown in Table 7. DEXI demonstrated the highest accuracy for diagnosing Nevi (NV), Basal Cell Carcinomas (BCC), Benign Keratosis-like Lesions (BKL), and Dermatofibromas (DF). The expert classifier excelled in identifying Melanoma (MEL), while DEXI and the expert performed comparably on Actinic Keratosis (AK) and Squamous Cell Carcinoma (SCC).

The non-expert’s diagnostic performance was generally lower across all lesion types, though their accuracy was relatively stronger on malignant lesion types than benign ones (Figure 2).

### 3.5. Performance Metrics

Precision, recall, and F1 scores were calculated for each classifier to further quantify diagnostic performance (Table 8). These were derived using micro-averaging, which sums class-wise true positives before computing each metric.

DEXI achieved the highest scores across all three metrics: precision (0.665), recall (0.634), and F1 score (0.632), reflecting a balanced and reliable performance. The expert scored 0.433, 0.458, and 0.387, respectively—moderate values indicating good but less consistent performance. The non-expert yielded the lowest scores across all metrics.

Statistical analysis using exact permutation testing confirmed these performance differences. In comparisons between DEXI and the expert, *p*-values were 0.0229 (precision), <0.001 (recall), and <0.001 (F1 score), indicating that DEXI significantly outperformed the expert across all measures.

Similarly, all comparisons between DEXI and the non-expert yielded *p*-values < 0.001. Comparisons between the expert and non-expert also revealed significant differences, with *p*-values of 0.0270 (precision), <0.001 (recall), and 0.0240 (F1 score).

## 4. Discussion

This study investigated the diagnostic performance of three classifiers—an AI model (DEXI), an expert dermatologist, and a non-expert medical student—in categorizing skin lesions and classifying them as benign or malignant. Using a comprehensive set of evaluation metrics, including accuracy, sensitivity, specificity, chi-square testing, and performance measures such as precision, recall, and F1 score, DEXI demonstrated the most effective overall performance. It achieved a sensitivity of 0.936 and a specificity of 0.704, highlighting its ability to correctly identify malignant lesions while also minimizing false positives.

In comparison, both human classifiers showed strong sensitivity but substantially lower specificity, indicating a greater tendency to misclassify benign lesions as malignant. This pattern suggests a conservative diagnostic approach among human evaluators—particularly the expert—who may err on the side of caution in ambiguous cases, potentially leading to increased false-positive rates. Notably, the non-expert, despite limited training, exhibited strong sensitivity in detecting malignancies, underscoring the nuances and variability of diagnostic expertise across experience levels.

DEXI showed strong diagnostic accuracy for lesion types such as nevi (83.33%), melanoma (85.18%), BCC (85.71%), and SCC (66.66%). However, its performance significantly declined for actinic keratosis (AK, 33.33%) and benign keratosis-like lesions (BKL, 34.61%). This suggests difficulty in distinguishing between AK and other benign lesions, potentially due to overlapping dermoscopic features such as pigmentation or scale. The reduced accuracy may reflect an insufficient representation of these lesion types in the AI training dataset or the intrinsic complexity of differentiating them. Additional training data and model refinement may be necessary to improve performance in these categories.

The expert classifier performed exceptionally well in diagnosing melanoma (94.44%) and BCC (80.95%), but was less effective for nevi (28.88%) and BKL (15.38%). Similarly, the non-expert showed higher accuracy for malignant lesions, such as melanoma (77.77%) and BCC (71.42%), while performing poorly for BKL (3.84%). These results illustrate how even experienced clinicians may struggle with visually similar benign lesions and reinforce the critical influence of both training and lesion type on diagnostic accuracy.

The consistent misclassification of AK and BKL across all classifiers further underscores the challenge these lesions present. Their visual similarity likely contributes to diagnostic confusion, and their apparent underrepresentation in training datasets may hinder both AI learning and human recognition. Improving classification performance will likely require increased inclusion of these lesion types in training data, along with enhancements in both algorithmic sensitivity and clinician education.

DEXI also outperformed both human classifiers in precision (0.665), recall (0.634), and F1 score (0.632), reflecting a more balanced diagnostic profile. Although the expert achieved higher accuracy in melanoma diagnosis, DEXI consistently surpassed both the expert and non-expert across overall performance metrics. Statistical comparisons confirmed these differences, with DEXI significantly outperforming the human classifiers in all core diagnostic measures. The performance gap between the expert and non-expert also emphasizes the impact of experience and training on diagnostic reliability.

The dataset used in this study, drawn from the University of Modena and Reggio Emilia, featured highly suspicious lesions primarily from Fitzpatrick skin types I–III. This clinically curated cohort does not represent the general population’s mole distribution but rather reflects lesions warranting excision or advanced assessment, due to the presence of atypical dermoscopic features that would prompt expert evaluation. The nature of this dataset likely contributed to the high sensitivity observed in all classifiers, particularly DEXI. Importantly, DEXI’s high specificity suggests potential clinical utility in minimizing unnecessary procedures, reducing patient anxiety, and lowering healthcare costs. However, the lower specificity of human classifiers highlights the need for additional confirmatory tests to address false positives.

As artificial intelligence becomes more integrated into dermatologic workflows, ethical and practical considerations must be addressed. Key among these are data privacy, algorithmic bias, and the risk of AI-driven misdiagnosis. Ensuring secure, de-identified storage of clinical images and associated metadata is essential to maintain patient confidentiality and public trust. Additionally, algorithmic bias can arise from underrepresented lesion types or demographic groups in training data. In this study, DEXI’s reduced performance on AK and BKL may reflect such gaps. Expanding datasets to include more diverse lesion types and populations would likely improve generalizability and fairness in AI-driven diagnostics.

Moreover, this study relied solely on 2D dermoscopic images, which may limit real-world applicability. Clinical assessment often incorporates 3D imaging or dynamic, temporal evaluations that provide richer contextual information. While 2D imaging remains a valuable tool, future AI models may benefit from incorporating multispectral or 3D modalities to enhance diagnostic accuracy and model robustness in varied clinical settings.

Human oversight remains indispensable. While AI systems like DEXI can significantly improve diagnostic efficiency and consistency, especially in routine screening contexts, they should complement—not replace—clinical judgment. Expert intervention is crucial in ambiguous or high-stakes cases, where misdiagnosis could result in inappropriate management. In practice, AI can serve as a preliminary screening aid, flagging high-risk lesions and enabling dermatologists to focus on complex presentations, thereby optimizing workflows and improving outcomes [60].

The observed performance gap between the expert and non-expert classifiers underscores the importance of structured training and educational support for novice practitioners. The non-expert in this study received short, informal training without a standardized curriculum or assessment, which likely limited their diagnostic precision. Future studies should incorporate formalized, feedback-driven training programs aligned with current diagnostic standards. These programs might combine theory (e.g., risk assessment frameworks and dermoscopy principles, as outlined in our “Diagnostic Process of Skin Lesions” chapter) with supervised practical experience and regular competency evaluations. A more structured approach would enable more accurate benchmarking and meaningful comparisons with AI and expert performance.

Previous literature has shown that AI models can match or exceed dermatologist performance in classifying pigmented lesions [61]. However, their diagnostic accuracy in real-world settings is often influenced by external factors such as case complexity, image quality, and available infrastructure. The findings from this study strengthen DEXI’s case for clinical adoption by demonstrating both high sensitivity and specificity—especially in a dataset enriched with diagnostically challenging cases. These performance metrics align with key criteria used in regulatory evaluations and support DEXI’s potential for deployment in clinical practice.

That said, successful clinical integration also depends on factors beyond diagnostic accuracy. Usability features such as intuitive interfaces, seamless electronic health record (EHR) integration, and fast image processing are critical for workflow adoption. In resource-limited settings, DEXI may be particularly valuable in supporting non-specialist clinicians. However, scalability will depend on addressing computational constraints, infrastructure gaps, and delivering tailored training for local healthcare providers. Ethical considerations—especially data privacy and mandatory human oversight—are also central to achieving regulatory approval and responsible use.

This study contributes meaningfully to the growing body of literature by comparing AI diagnostic performance with human classifiers of differing experience levels. Additionally, it highlights the impact of training, dataset diversity, and real-world clinical contexts on classification accuracy. The use of a clinically curated dataset from an Italian patient population further enhances the study’s value by evaluating AI performance in a demographic group less commonly represented in datasets like ISIC, which are primarily sourced from institutions in the United States, Australia, Spain, Austria, and Argentina.

By including multiple images per lesion—even in 2D—this study offers a nuanced evaluation of diagnostic performance, reflecting real-world variability in lesion presentation. Future research should focus on strategies to reduce performance disparities between classifiers. These may include additional training for human practitioners, the integration of decision support systems, or further refinement of AI models and training datasets, particularly in settings where lesions are clinically suspected to be malignant.

In summary, DEXI has demonstrated similar sensitivity to human classifiers, while maintaining a notably higher specificity. The difference in classifications between the human readers and the algorithm, however, emphasizes the importance of human expertise in dermatological diagnosis. The findings advocate for a combined approach, where AI serves as a valuable tool to augment human judgment, particularly in complex cases. Continued research and development are essential to fully harness the potential of AI in dermatology, ensuring that these tools can be effectively integrated into clinical practice to improve patient outcomes.

## 5. Conclusions

This study evaluated the diagnostic performance of an artificial intelligence model (DEXI), an expert dermatologist, and a non-expert in classifying skin lesions as benign or malignant using dermoscopic images. Across a range of performance metrics—including sensitivity, specificity, Cohen’s kappa, precision, recall, F1 score, and lesion-type accuracy—DEXI consistently outperformed both human classifiers, particularly in its ability to correctly identify benign lesions and reduce false positives. While the expert dermatologist exhibited high sensitivity and excelled in diagnosing melanoma, DEXI demonstrated a more balanced overall performance, making it a strong candidate for integration into clinical workflows.

The findings underscore the potential of AI to support diagnostic decision-making in dermatology, particularly in high-volume or resource-limited settings. However, consistent misclassification of certain lesion types, such as actinic keratosis and benign keratosis-like lesions, suggests the need for improved training datasets and algorithmic refinement. Furthermore, the observed performance gap between expert and non-expert classifiers highlights the critical role of structured training and ongoing education in achieving diagnostic consistency among clinicians.

Ethical considerations—including data privacy, algorithmic bias, and the necessity of human oversight—remain central to the clinical deployment of AI tools. For AI models like DEXI to be effectively and responsibly adopted, they must not only meet regulatory standards for diagnostic accuracy but also demonstrate usability, fairness, and seamless integration into existing clinical infrastructures.

By leveraging a dataset of histologically confirmed lesions from an Italian patient population, this study adds important geographic and demographic diversity to the field of AI dermatology. The inclusion of multiple images per lesion further enriches the analysis, providing a closer approximation of real-world diagnostic conditions. Taken together, these contributions strengthen the case for a collaborative approach in dermatological diagnostics, where AI serves as a valuable tool to augment, rather than replace, clinical expertise.

Future research should focus on enhancing AI model generalizability through expanded datasets, integrating diverse imaging modalities, and developing hybrid diagnostic systems that combine human judgment with algorithmic support. With continued development and ethical implementation, AI has the potential to improve diagnostic accuracy, reduce healthcare disparities, and support better outcomes in dermatologic care.

## Figures and Tables

**Figure 1 diagnostics-15-01115-f001:**
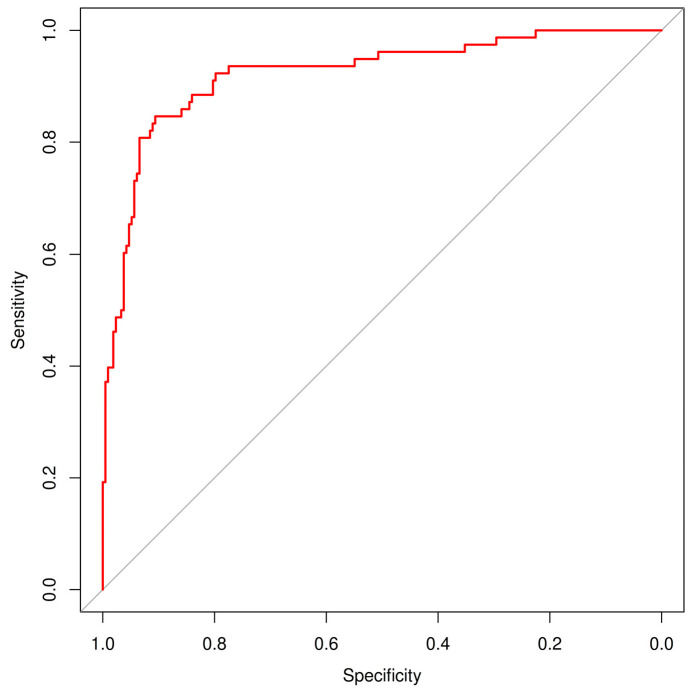
Area under the ROC curve (AUC) for AI (DEXI).

**Figure 2 diagnostics-15-01115-f002:**
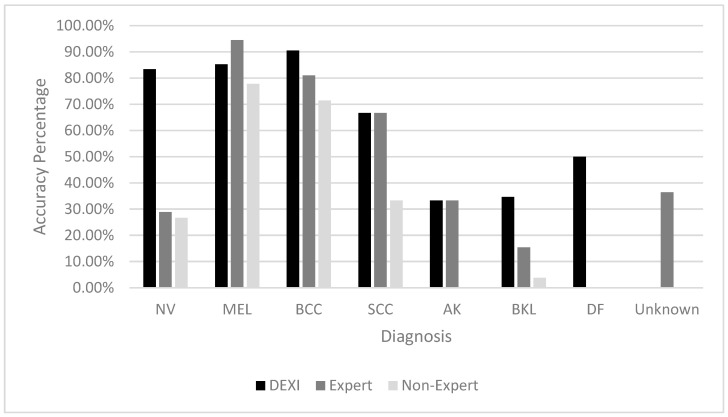
Distribution of accurate diagnosis per classifier.

**Table 1 diagnostics-15-01115-t001:** ISIC archive diagnostic lesions.

Lesion	Benign/Malignant
Actinic Keratosis	Benign/Malignant
Angiofibroma or Fibrous Papule	Benign
Angioma	Benign
Atypical Melanocytic Proliferation	Bening/Malignant
Basal Cell Carcinoma	Malignant
Café-Au-Lait Macules	Benign
Dermatofibroma	Benign
Lentigo NOS	Benign/Malignant
Lentigo Simplex	Benign
Lichenoid Keratosis	Benign
Melanoma	Malignant
Nevus	Benign
Pigmented Benign Keratosis	Benign
Scar	Benign/Malignant
Seborrheic Keratosis	Benign
Solar Lentigo	Benign
Squamous Cell Carcinoma	Malignant
Vascular Lesion	Benign/Malignant
Unknown	Benign/Malignant

**Table 2 diagnostics-15-01115-t002:** Ground truth diagnostic output.

Lesion	Benign/Malignant
NV (Nevus)	Benign
MEL (Melanoma)	Malignant
BCC (Basal Cell Carcinoma)	Malignant
SCC (Squamous Cell Carcinoma)	Malignant
AK (Actinic Keratosis)	Benign
BKL (Benign Keratosis, e.g., Lentigines, Seborrheic Keratosis, Lichenoid Keratosis)	Benign
DF (Dermatofibroma)	Benign
VASC (Vascular Lesions)	Benign

**Table 3 diagnostics-15-01115-t003:** Correct lesion benign/malignant classification for each classifier.

Classifier	Benign	Malignant	Total Correct Guesses
AI (DEXI)	150 (70.42%)	73 (93.59%)	223
Expert	70 (32.86%)	73 (93.59%)	143
Non-Expert	61 (28.64%)	74 (94.87%)	135
Ground Truth (N)	213	78	-

**Table 4 diagnostics-15-01115-t004:** Confusion matrix for lesion classification.

Classifier	True Negatives	True Positives	False Positives	False Negative
AI (DEXI)	150	73	63	5
Expert	70	73	143	5
Non-Expert	61	74	152	4

**Table 5 diagnostics-15-01115-t005:** Sensitivity and specificity for each classifier.

Classifier	Sensitivity	Specificity
AI (DEXI)	0.936	0.704
Expert	0.936	0.328
Non-Expert	0.948	0.286

**Table 6 diagnostics-15-01115-t006:** Inter-rater agreement: Cohen’s kappa values.

Comparison	Kappa	95% Confidence Interval
AI (DEXI) vs. Expert	0.23	0.14–0.32
AI (DEXI) vs. Non-Expert	0.28	0.19,0.36
Expert vs. Non-Expert	0.43	0.31,0.55

**Table 7 diagnostics-15-01115-t007:** Accurate lesion diagnosis by type.

Classifier	NV	MEL	BCC	SCC	AK	BKL	DF	Unknown	Total
AI (DEXI)	150 (83.33%)	46 (85.18%)	19 (90.47%)	2 (66.66%)	1 (33.33%)	9 (34.61%)	2 (50%)	0	229
Expert	52 (28.88%)	51 (94.44%)	17 (80.95%)	2 (66.66%)	1 (33.33%)	4 (15.38%)	0	4 (36.36%)	127
Non-Expert	48 (26.66%)	42 (77.77%)	15 (71.42%)	1 (33.33%)	0	1 (3.84%)	0	0	107
Ground Truth (N)	180	54	21	3	3	26	4	11	302

**Table 8 diagnostics-15-01115-t008:** Diagnostic Performance Metrics.

Classifier	Precision	Recall	F1 Score
AI (DEXI)	0.665	0.634	0.632
Expert	0.433	0.458	0.387
Non-Expert	0.270	0.305	0.285

## Data Availability

The datasets presented in this article are not readily available because the data are part of an ongoing study. Requests to access the datasets should be directed to carmen.cantisani@uniroma1.it.

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
