# Peer review of "A Comparison of Skin Lesions’ Diagnoses Between AI-Based Image Classification, an Expert Dermatologist, and a Non-Expert"

_diagnostics, 2025, doi:10.3390/diagnostics15091115_

Round 1

Reviewer 1 Report (Previous Reviewer 1)

Comments and Suggestions for Authors

Authors have modified the article as per review comments. Accept the article.

Author Response

We thank the reviewer for their positive feedback and for recommending acceptance. We are grateful for the thoughtful comments and suggestions that helped improve the quality of the manuscript.

Reviewer 2 Report (Previous Reviewer 2)

Comments and Suggestions for Authors

The author's clarifications have effectively addressed all concerns.

Author Response

We thank the reviewer for their positive feedback and for recommending acceptance. We are grateful for the thoughtful comments and suggestions that helped improve the quality of the manuscript.

Reviewer 3 Report (Previous Reviewer 4)

Comments and Suggestions for Authors

Review Report for MDPI Diagnostics
(A comparison of skin lesions' diagnoses between AI-based image classification, an expert dermatologist and a non-expert)

1.    Within the scope of this study, the diagnostic accuracy of an AI-based image classification system was compared with an expert dermatologist and a non-expert in classifying skin lesions, demonstrating that while AI can assist dermatologists, it is not yet a replacement for expert clinical judgment.

2.    In the introduction section, the types of skin tumors, dermoscopy, epidemiology, diagnostic processes related to skin lesions, the importance of the subject and the place of artificial intelligence in dermatology are mentioned to a certain extent. In this section, it is suggested to add a table consisting of sections such as "dataset used, method, originality, pros and cons" for the literature in the place of artificial intelligence in dermatology section, especially in order to present the literature more clearly.

3.    Although the purpose of the study has been clearly stated, the differences of the study from the literature and its main contributions to the literature should be added in bullet points and more emphatically at the end of this section and just before the "materials and methods" section.

4.    The dataset used in the study seems to be at a sufficient level when considered in terms of the subject of the study and the amount and type of data. However, it is suggested to add the superiority and/or differences of the dataset used in the study compared to other open source datasets in the literature in this section.

5.    In this study, where AI, Expert and Non-Expert are expressed as classifiers, the details of the preferred model, its architecture and originality section should be included, especially in relation to AI. The amounts such as training, validation, testing in the amount of data and information such as the hardware and software used, toolbox should be detailed.

6.    Although a certain level of metric type is included in terms of performance metrics, it is suggested that Cohen's cappa score, receiver operating characteristic (ROC) curve, AUC (area under the ROC curve) score metrics should be included in relation to the AI model, if possible.

As a result, the study is interesting in terms of subject, but attention should be paid to the sections mentioned above.

Comments on the Quality of English Language

Quality of English Language is at a certain level. However, the pdf of the paper has been uploaded with track changes open.

Author Response

Reviewer Comment 2:

“In the introduction section, the types of skin tumors, dermoscopy, epidemiology, diagnostic processes related to skin lesions, the importance of the subject and the place of artificial intelligence in dermatology are mentioned to a certain extent. In this section, it is suggested to add a table consisting of sections such as ‘dataset used, method, originality, pros and cons’ for the literature in the place of artificial intelligence in dermatology section, especially in order to present the literature more clearly.”

Response: We sincerely thank the reviewer for this excellent suggestion. We initially attempted to compile a comparative table summarizing the literature on AI algorithms in dermatological diagnostics, including sections such as dataset used, methodology, originality, and pros and cons. However, upon further consideration, we realized that the scope of the literature cited in our manuscript is quite broad, and adequately summarizing the diverse range of AI algorithms and their respective advantages and limitations would require a more extensive review than a single table could accommodate.

If the reviewer feels it would strengthen the manuscript, we would be glad to expand this section into a focused literature review. In that case, we believe a comparison of AI models with human classifiers may be more relevant to our study's objectives, as our work centers on diagnostic accuracy among classifiers rather than technical evaluation of individual AI model architectures—particularly given that the internal workings of the DEXI algorithm used in our study are proprietary and not publicly available.

We are open to incorporating a more targeted review should the reviewer consider it beneficial for contextualizing our contribution.

Reviewer Comment 3:

“Although the purpose of the study has been clearly stated, the differences of the study from the literature and its main contributions to the literature should be added in bullet points and more emphatically at the end of this section and just before the ‘materials and methods’ section.”

Response: Thank you for this valuable recommendation. As requested, we have added a bullet-pointed summary at the end of the Introduction and before the Materials and Methods section to clearly highlight the main contributions of our study. These points emphasize:

  • Independent evaluation of the proprietary AI model DEXI on a previously unseen dataset from an Italian population.
  • Direct comparison of DEXI with both an expert dermatologist and a non-expert (medical student), offering a unique three-way assessment.
  • Use of real-world dermoscopic images from a high-risk clinical setting to evaluate diagnostic performance in practical scenarios.

Reviewer Comment 4:

“The dataset used in the study seems to be at a sufficient level when considered in terms of the subject of the study and the amount and type of data. However, it is suggested to add the superiority and/or differences of the dataset used in the study compared to other open source datasets in the literature in this section.”

Response: We appreciate the reviewer’s suggestion. We have now elaborated on the strengths of the dataset in the Materials and Methods section. Compared to publicly available datasets like ISIC, our dataset:

  • Consists of images with histopathological confirmation.
  • Reflects real-world clinical decision-making, as all lesions were selected due to clinical suspicion of malignancy.
  • Represents a distinct geographic population (Italian patients), enhancing diversity beyond the ISIC’s usual demographic base (e.g., U.S., Australia, Spain).
  • Includes multiple images per lesion, allowing for a richer assessment of classifier performance.

Reviewer Comment 5:

“In this study, where AI, Expert and Non-Expert are expressed as classifiers, the details of the preferred model, its architecture and originality section should be included, especially in relation to AI. The amounts such as training, validation, testing in the amount of data and information such as the hardware and software used, toolbox should be detailed.”

Response: Thank you for this important point. We have expanded the Materials and Methods section to include available details on the AI model (DEXI):

  • DEXI is based on the EfficientNetV2 CNN architecture and outputs malignancy risk scores (0–10 scale).
  • It was trained on both proprietary datasets and publicly available images from the ISIC archive (81,155 images).
  • Due to its proprietary nature, detailed architecture, hyperparameters, and software environment were not disclosed to us by the developers.
  • However, we clarified the version used (DEXI 2.1) and elaborated on how predictions were made (average malignancy score across all images per lesion).
  • We have also indicated that the classification task was evaluated on a dataset of 302 histopathologically confirmed lesions (1,047 images), with performance metrics such as sensitivity, specificity, and F1 score reported.

Reviewer Comment 6:

“Although a certain level of metric type is included in terms of performance metrics, it is suggested that Cohen's kappa score, receiver operating characteristic (ROC) curve, AUC (area under the ROC curve) score metrics should be included in relation to the AI model, if possible.”

Response: We acknowledge and appreciate the reviewer’s suggestion to include additional metrics such as the Cohen’s kappa score, ROC curves, and AUC. In consultation with our biostatistician, we initially prioritized metrics such as sensitivity, specificity, precision, recall, and F1 score to evaluate diagnostic performance.

However, in response to this comment, our biostatistician will provide the results of the Cohen’s kappa score and ROC curves with the corresponding AUC values upon his return from vacation. These updated analyses will be included in the revised manuscript or supplementary material, where appropriate.

This manuscript is a resubmission of an earlier submission. The following is a list of the peer review reports and author responses from that submission.

Round 1

Reviewer 1 Report

Comments and Suggestions for Authors

How does the study ensure the reproducibility of the DEXI classifier's results using the dataset and methodology described?

What measures were taken to address potential biases in the dataset used for training and validation?

Can author elaborate on why certain lesions with ambiguous characteristics were predetermined as benign or malignant in the ground truth?

Could author clarify the rationale for the 3.5 cut-off value in DEXI's malignancy score classification?

How do author propose integrating AI classifiers like DEXI into clinical workflows without undermining the clinical judgment of dermatologists?

What are the limitations of the study in terms of generalizing the findings to diverse populations or lesion types?

How would the study’s conclusions change if a more balanced dataset with equal representation of lesion types was used?

How scalable is DEXI’s methodology for use in resource-constrained settings or low-resource clinics?

Could author provide a comparison of the time efficiency of diagnosis between DEXI and human evaluators?

Did the study evaluate user feedback from the expert and non-expert participants regarding the usability of DEXI?

Could author discuss the potential implications of author findings for regulatory approvals and clinical adoption of AI-based diagnostic tools?

Could author provide additional details on the dataset characteristics, particularly regarding its representation of diverse demographic groups and Fitzpatrick skin types?

How do author plan to address the misclassification challenges observed with AK and BKL lesions in future AI model training?

Can author elaborate on the structured training approach author propose for non-experts to improve their diagnostic accuracy?

What specific steps will author take to mitigate algorithmic biases in AI models when applied to more diverse populations?

Could author clarify how the inclusion of 2D images might impact the generalizability of the findings to real-world diagnostic settings?

What mechanisms are in place to ensure data privacy and compliance with ethical standards in author study?

How do author interpret the chi-square analysis results in the context of practical applications for clinical diagnostics?

Can author explain why DEXI struggles with certain lesion types and how this affects its clinical reliability?

What implications do the differences in performance between expert and non-expert classifiers have for future training programs in dermatology?

How does the variability in lesion classification between AI and human classifiers impact author proposed integration of AI into clinical practice?

Can author provide specific examples of cases where human oversight significantly improved AI's diagnostic decisions?

Could author discuss how expanding the dataset might improve DEXI’s specificity for benign lesions?

How do author propose balancing the benefits of AI-driven diagnostics with the ethical concerns surrounding their deployment?

How might the findings from this study translate into improved clinical workflows for skin lesion diagnostics?

Comments on the Quality of English Language

The English could be improved to more clearly express the research.

Author Response

How does the study ensure the reproducibility of the DEXI classifier's results using the dataset and methodology described?

We acknowledge the reviewer’s concerns regarding reproducibility. The DEXI algorithm was trained on proprietary datasets from Canfield Scientific as well as publicly available sources like the ISIC archive. Due to the proprietary nature of these datasets and the algorithm, we are unable to provide complete details or ensure full reproducibility. The testing data, obtained from Modena University, is also under restrictions due to ongoing research. Our study's objective is to assess DEXI’s diagnostic utility rather than its internal architecture. To address concerns about reproducibility, we have clearly documented the methodologies employed, including the dataset characteristics and evaluation metrics, to facilitate transparency. Future studies could benefit from open-source AI models and publicly available datasets to enhance reproducibility.

What measures were taken to address potential biases in the dataset used for training and validation?

We recognize the limitations posed by biases in the datasets used for training and validation. These biases may include the underrepresentation of certain lesion types, Fitzpatrick skin types IV-VI, and rare conditions. The training dataset predominantly consisted of Fitzpatrick skin types I-III, which limits the generalizability of the findings to more diverse populations. While we had no control over the composition of these datasets, we have discussed these biases and their implications in the manuscript. Future research should prioritize curating datasets that include a more balanced representation of lesion types and skin tones to mitigate these biases and improve AI model performance across diverse populations.

Can author elaborate on why certain lesions with ambiguous characteristics were predetermined as benign or malignant in the ground truth?

The predetermined classification of ambiguous lesions as benign or malignant in the ground truth was based on their most common clinical presentations, which served as a practical and consistent approach given the limitations of the dataset. The dataset from Modena provided only lesion diagnoses without further classifications, necessitating a standardized framework to categorize lesions for analysis. For example, actinic keratosis (AK), while clinically recognized as a precancerous condition due to its potential progression to squamous cell carcinoma (SCC), is most commonly categorized as benign in dermatological practice. Similarly, nevi are predominantly benign, although a small subset may progress to melanoma. These predetermined classifications were intended to align with established dermatological norms and ensure consistency across all classifiers. By standardizing classifications based on common clinical presentations, the authors aimed to mitigate variability in how ambiguous lesions were interpreted by human evaluators and DEXI, facilitating a comparative statistical analysis. This approach was essential to minimize inconsistencies within the study.

Could author clarify the rationale for the 3.5 cut-off value in DEXI's malignancy score classification?

The 3.5 cut-off value for malignancy classification is a proprietary threshold set by Canfield Scientific, based on their algorithm’s training and validation. As we are not directly involved in the development of DEXI, we are unable to provide further details about how this threshold was determined. We suggest that future studies collaborate with the developers of proprietary tools to provide greater transparency around these parameters.

How do author propose integrating AI classifiers like DEXI into clinical workflows without undermining the clinical judgment of dermatologists?

We propose that AI tools like DEXI be integrated as adjunct systems to support dermatologists rather than replace their expertise. AI can serve as a second opinion, offering consistent and efficient preliminary assessments, particularly for routine screenings. Dermatologists can then focus on complex cases requiring nuanced clinical judgment. This integration can streamline workflows, reduce diagnostic errors, and enhance patient outcomes. We recommend pilot studies to evaluate the practical implementation of AI systems in various clinical settings.

What are the limitations of the study in terms of generalizing the findings to diverse populations or lesion types?

We acknowledge that the study’s conclusions might differ if a more balanced dataset were used. For instance, equal representation of lesion types could enhance the AI’s ability to correctly identify underrepresented categories, such as benign keratosis-like lesions (BKL). Future research should involve datasets with proportional representation of all lesion types and demographic groups to provide a more comprehensive evaluation of AI diagnostic capabilities.

How would the study’s conclusions change if a more balanced dataset with equal representation of lesion types was used?

A more balanced dataset with equal representation of lesion types could improve the study's conclusions by enhancing the performance of all classifiers, particularly for underrepresented lesions like actinic keratosis (AK) and benign keratosis-like lesions (BKL), where DEXI struggled. This would reduce algorithmic biases, improve the generalizability of findings to real-world clinical scenarios, and provide a more equitable basis for evaluating the comparative performance of AI, experts, and non-experts. It could also offer deeper insights into specificity improvements, particularly for benign lesions, resulting in a more comprehensive assessment of AI's diagnostic capabilities.

How scalable is DEXI’s methodology for use in resource-constrained settings or low-resource clinics?

As I am not the developer of DEXI, I cannot confirm its scalability. However, based on the study, DEXI shows potential for use in resource-constrained settings by assisting in the evaluation of skin lesions, which could help non-specialists identify high-risk lesions. To assess its scalability, factors such as infrastructure needs, simplified computational requirements, and training for local healthcare providers would need to be addressed in low-resource environments.

Could author provide a comparison of the time efficiency of diagnosis between DEXI and human evaluators?

Based on our estimates, human evaluators required approximately three hours to diagnose 1,000 images, whereas DEXI generated results in mere seconds. This significant difference highlights DEXI’s potential to reduce diagnostic turnaround times, enabling quicker decision-making and improved efficiency in dermatological workflows. Future studies could systematically compare diagnostic time efficiency across various AI tools and human evaluators.

Did the study evaluate user feedback from the expert and non-expert participants regarding the usability of DEXI?

The study did not evaluate user feedback from the expert and non-expert participants regarding the usability of DEXI, as they did not use the AI model directly. Instead, their diagnostic accuracy was assessed independently and compared to the performance of DEXI.

Could author discuss the potential implications of author findings for regulatory approvals and clinical adoption of AI-based diagnostic tools?

The findings of this study have important implications for the regulatory approval and clinical adoption of AI-based diagnostic tools, particularly in dermatology. DEXI’s demonstrated high sensitivity and specificity in detecting malignant lesions, especially with fewer false positives, make it a promising tool for improving diagnostic accuracy. This aligns well with key regulatory criteria for AI-based diagnostic tools, such as clinical effectiveness and the reduction of diagnostic errors. However, while DEXI shows promise for use in resource-constrained settings, where non-specialists can assist in identifying high-risk lesions, its scalability will require addressing challenges like infrastructure limitations, simplified computational requirements, algorithmic biases, and training local healthcare providers. Furthermore, ethical considerations, such as ensuring data privacy and incorporating human oversight to mitigate the risk of misdiagnosis, are essential for gaining regulatory approval and ensuring safe, equitable clinical integration.

Could author provide additional details on the dataset characteristics, particularly regarding its representation of diverse demographic groups and Fitzpatrick skin types?

The datasets used in this study predominantly represented Fitzpatrick skin types I-III, limiting the findings’ applicability to populations with darker skin tones. This limitation underscores the need for more inclusive datasets in future research to ensure AI tools perform well across diverse demographic groups. Expanding dataset diversity will also help address algorithmic biases and improve diagnostic equity.

How do author plan to address the misclassification challenges observed with AK and BKL lesions in future AI model training?

The authors of this article are not AI engineers, so addressing the technical challenges of AI model training is beyond their expertise. However, to improve the misclassification of actinic keratoses (AK) and benign keratoses (BKL) lesions, they suggest strategies such as incorporating larger, more diverse datasets, leveraging expert-annotated training sets, and integrating patient metadata with dermoscopic images. These approaches could help guide AI engineers in refining model performance for these challenging lesion types.

Can author elaborate on the structured training approach author propose for non-experts to improve their diagnostic accuracy?

We propose a structured training regimen for non-experts, combining theoretical knowledge with practical exposure and real-time feedback. This curriculum could include modules from our “Diagnostic Process of Skin Lesions” chapter, supplemented by diagnostic dermatology textbooks and dermoscopic atlases. Regular assessments and supervised clinical practice would help improve diagnostic accuracy over time. Implementing this approach could bridge the gap between non-expert and expert performance.

What specific steps will author take to mitigate algorithmic biases in AI models when applied to more diverse populations?

To mitigate algorithmic biases in AI models when applied to more diverse populations, the authors propose several key steps. First, expanding the training datasets to include a more diverse range of demographic groups and skin types is crucial. This would help reduce biases that arise from underrepresentation, particularly of lesions like actinic keratoses (AK) and benign keratoses (BKL), which the AI model struggled with in this study. A more inclusive dataset will improve the model’s ability to generalize across different populations, ensuring more accurate diagnoses for individuals from varying backgrounds. Additionally, continuous monitoring and refinement of AI models based on real-world data can help identify and address any emerging biases. By improving the diversity of training data and ensuring regular updates, the AI model’s diagnostic performance can be enhanced, making it more equitable and applicable to a broader range of clinical settings.

Could author clarify how the inclusion of 2D images might impact the generalizability of the findings to real-world diagnostic settings?

The inclusion of 2D images in this study may limit the generalizability of the findings to real-world diagnostic settings, where 3D imaging or more complex imaging techniques are often used. While 2D images provide valuable insights into lesion characteristics, they may not capture the full depth or complexity of a lesion, which could impact the accuracy of diagnoses in clinical practice. In real-world settings, dermatologists often rely on 3D imaging or dynamic assessment of lesions, which can offer more detailed information about their structure and behavior over time. The study's reliance on 2D images could therefore lead to some limitations in the AI model's ability to generalize to more varied clinical environments that use different imaging technologies. To better align with real-world applications, future research could include a broader range of imaging modalities, such as 3D or multispectral images, to further assess the AI model’s performance and enhance its generalizability.

What mechanisms are in place to ensure data privacy and compliance with ethical standards in author study?

Data privacy and compliance with ethical standards in this study were ensured by the organizations that provided the data: the ISIC archive, Canfield Scientific, and Modena University. These institutions implemented mechanisms to protect patient confidentiality, including secure data storage and de-identification of images and metadata. By adhering to established ethical guidelines and privacy protocols, they ensured the responsible handling of all data used in the study.

How do author interpret the chi-square analysis results in the context of practical applications for clinical diagnostics?

The significant chi-square results underscore the performance disparities among classifiers. These findings emphasize the importance of combining AI tools with human expertise to optimize diagnostic accuracy. AI’s consistency and efficiency can complement dermatologists’ nuanced clinical judgment, creating a synergistic approach to skin lesion diagnostics.

Can author explain why DEXI struggles with certain lesion types and how this affects its clinical reliability?

DEXI struggles with certain lesion types, particularly AK and BKL, due to overlapping visual characteristics such as atypical pigmentation or size. These shared features make accurate differentiation challenging, which affects the model’s clinical reliability for these lesion types. Additionally, the high misclassification rates for AK and BKL suggest that these lesions might be underrepresented in the training dataset. This limitation impacts DEXI’s ability to generalize effectively in clinical settings, where accurate identification of these lesions is critical for appropriate diagnosis and treatment. Addressing this challenge is essential to enhance the model's performance and its applicability in real-world scenarios.

What implications do the differences in performance between expert and non-expert classifiers have for future training programs in dermatology?

The differences in performance between expert and non-expert classifiers highlight the critical role of comprehensive and structured training programs in dermatology. Experts demonstrated superior accuracy in diagnosing complex lesion types such as melanoma (MEL), underscoring the importance of in-depth knowledge and clinical experience. In contrast, non-experts, despite reasonable proficiency in some areas, exhibited significant limitations, particularly with benign lesions like BKL, due to their brief and less structured training. These findings suggest that future training programs should incorporate detailed theoretical knowledge alongside extensive practical experience, emphasizing lesion variability and diagnostic complexity. Regular competency assessments, feedback mechanisms, and adherence to standardized diagnostic guidelines can help bridge the gap between non-expert and expert performance. Such structured curricula can ensure that practitioners develop the diagnostic accuracy needed to complement advanced tools like AI while maintaining consistency and reliability in clinical practice.

How does the variability in lesion classification between AI and human classifiers impact author proposed integration of AI into clinical practice?

The variability in lesion classification between AI and human classifiers underscores both the advantages and challenges of integrating AI into clinical practice. While the AI model, DEXI, demonstrated higher specificity and balanced diagnostic performance, reducing false positives and streamlining workflows, its struggles with certain lesion types like AK and BKL highlight the need for improved training datasets and fine-tuning. Human classifiers, particularly experts, excelled in diagnosing complex lesions such as melanoma but showed greater variability in specificity and consistency. This emphasizes the importance of human oversight to address AI limitations and ensure accurate diagnoses in challenging cases. Integrating AI into clinical practice should adopt a complementary approach, leveraging AI for routine screening and high-risk lesion identification while allowing dermatologists to focus on complex and nuanced cases. This collaborative model can enhance diagnostic accuracy, improve patient outcomes, and ensure that AI augments rather than replaces human expertise.

Can author provide specific examples of cases where human oversight significantly improved AI's diagnostic decisions?

Human oversight is not necessarily expected to improve AI’s diagnostic decisions but rather to mitigate its errors. This is particularly critical in cases involving lesion types like AK and BKL, where DEXI struggled with misclassifications due to overlapping features. Expert dermatologists, with their nuanced clinical judgment, are better equipped to identify subtle patterns and contextual information that AI might miss. For instance, while DEXI excelled in detecting melanoma with high sensitivity, human oversight can address the limitations of AI in ambiguous cases, ensuring accurate diagnoses and preventing misdiagnosis or inappropriate treatment. This underscores the role of human expertise in complementing AI, particularly in complex or high-stakes clinical scenarios.

Could author discuss how expanding the dataset might improve DEXI’s specificity for benign lesions?

Expanding the dataset could significantly improve DEXI’s specificity for benign lesions by addressing underrepresentation and enhancing its ability to differentiate between visually similar lesion types. The study highlights that DEXI struggles with lesions like AK and BKL, which share overlapping features such as atypical pigmentation and size. These challenges may stem from a lack of diverse examples of these lesion types in the training data. By incorporating a broader range of benign lesions and demographic groups into the dataset, the model could better learn the subtle distinctions between lesion types, reducing false-positive rates and improving specificity. Furthermore, including advanced imaging modalities, such as 3D or multispectral imaging, could provide richer data for training, enabling DEXI to capture the depth and complexity of lesion characteristics. Expanding the dataset in this way would not only enhance DEXI’s performance but also ensure its diagnostic capabilities are more generalizable and equitable across diverse clinical settings.

How do author propose balancing the benefits of AI-driven diagnostics with the ethical concerns surrounding their deployment?

The authors propose balancing the benefits of AI-driven diagnostics with ethical concerns by emphasizing data privacy, addressing algorithmic biases, and integrating human oversight into diagnostic workflows. To ensure patient confidentiality, they advocate for secure storage and de-identification of images and metadata. Algorithmic biases, which may arise from underrepresented demographic groups or lesion types in training datasets, can be mitigated by expanding and diversifying datasets to include broader lesion types and advanced imaging modalities. This would improve diagnostic equity and accuracy across diverse populations. Furthermore, human oversight is crucial to mitigate the risks of AI errors, which could lead to misdiagnoses and inappropriate treatments. AI should complement rather than replace human judgment, enhancing efficiency in routine screenings while allowing dermatologists to focus on complex cases. The authors also highlight the need for regulatory frameworks to ensure ethical deployment, addressing concerns such as infrastructure limitations and the need for training local healthcare providers. This balanced approach seeks to maximize AI’s potential while safeguarding ethical standards in clinical practice.

How might the findings from this study translate into improved clinical workflows for skin lesion diagnostics?

The findings from this study suggest several ways to improve clinical workflows for skin lesion diagnostics by integrating AI, such as DEXI, into current practices. DEXI’s high sensitivity and specificity, particularly in detecting malignant lesions, can streamline the initial screening process by efficiently identifying high-risk cases. This allows dermatologists to focus their expertise on complex or ambiguous lesions, reducing workload and enhancing diagnostic precision. The model’s ability to reduce false positives compared to human classifiers can minimize unnecessary follow-ups and patient anxiety, optimizing resource utilization. Additionally, incorporating AI into workflows can provide consistent and reproducible preliminary assessments, particularly in settings with limited access to dermatology experts.

To ensure effective integration, the study underscores the importance of human oversight to address AI errors and ethical concerns such as misdiagnosis risks and algorithmic biases. Expanding training datasets to include a broader range of lesions and demographics would further enhance the model’s performance and generalizability, particularly for underrepresented lesion types like AK and BKL. By complementing dermatologist expertise with AI’s efficiency, clinical workflows can become more accurate, equitable, and resource-efficient, ultimately improving patient outcomes.

Reviewer 2 Report

Comments and Suggestions for Authors

In the article entitled " A comparison of skin lesions' diagnoses between AI-based image classification, an expert dermatologist and a non-expert", the authors evaluate and compare the AI-based image classification of skin lesions to other methods of classification, an expert dermatologist, and a non-expert. The paper is interesting, especially considering the increasing use of AI in the health system. The article is well-written and easy to read with a good level of theoretical details.

There are only a few minor comments that may be considered:

While the study primarily focuses on sensitivity and specificity, it does not explore the user-friendliness of AI-based image classification. It would be beneficial to understand how easily AI can be integrated into existing dermatological workflows or systems. The authors may wish to expand on this topic.

AI-based image classification appears to be based on the assumption that consistent lighting and camera quality are key factors. However, variability in real-world settings could potentially impact the system's performance. It would be beneficial for the authors to provide more detailed information regarding the database of 15,178 images obtained from the dermatology department of the University of Modena and Reggio Emilia.

Author Response

In the article entitled "A comparison of skin lesions' diagnoses between AI-based image classification, an expert dermatologist and a non-expert", the authors evaluate and compare the AI-based image classification of skin lesions to other methods of classification, an expert dermatologist, and a non-expert. The paper is interesting, especially considering the increasing use of AI in the health system. The article is well-written and easy to read with a good level of theoretical details.

There are only a few minor comments that may be considered:

While the study primarily focuses on sensitivity and specificity, it does not explore the user-friendliness of AI-based image classification. It would be beneficial to understand how easily AI can be integrated into existing dermatological workflows or systems. The authors may wish to expand on this topic.

The study highlights the diagnostic accuracy of the AI model DEXI, focusing on metrics such as sensitivity and specificity, but does not directly address the user-friendliness of the AI system or its integration into existing dermatological workflows. Exploring this aspect would indeed be valuable, as the practical implementation of AI depends on its ease of use, compatibility with existing clinical systems, and the learning curve for practitioners. Factors such as intuitive interfaces, seamless integration with electronic health records (EHRs), and efficient image processing capabilities are crucial for ensuring widespread adoption. Additionally, the study could consider how the AI system fits into various clinical settings, from resource-rich hospitals to remote or under-resourced clinics. Addressing these considerations would provide a more comprehensive understanding of DEXI’s potential to improve dermatological workflows and its accessibility for practitioners with varying levels of technological proficiency. However, as the authors did not develop or directly test DEXI, but merely received the relevant data from Canfield Scientific, further elaboration on these aspects is beyond the scope of this manuscript.

AI-based image classification appears to be based on the assumption that consistent lighting and camera quality are key factors. However, variability in real-world settings could potentially impact the system's performance. It would be beneficial for the authors to provide more detailed information regarding the database of 15,178 images obtained from the dermatology department of the University of Modena and Reggio Emilia.

The dataset of 15,178 images provided by the dermatology department of the University of Modena and Reggio Emilia was integral to this study. It consisted of images of lesions excised based on clinical, dermoscopic, or confocal microscopy suspicions of melanoma or malignancy, resulting in a selection bias toward lesions with a high likelihood of malignancy. Furthermore, the dataset predominantly included Fitzpatrick skin types I–III, with limited representation of other skin types, which may have contributed to lower performance in certain lesion categories. Variability in real-world conditions, such as lighting and camera quality, could also affect the system's performance. Since the dataset was provided as-is, we did not have control over imaging conditions. However, as this proprietary data has been used in numerous esteemed publications, it was likely standardized in terms of lighting, camera settings, and lesion coverage. Each image depicted a portion of the lesion, with randomization applied during analysis using random.org. Although the database cannot be fully published due to ongoing research restrictions, it remains available for specific research purposes.

Reviewer 3 Report

Comments and Suggestions for Authors

The introduction would benefit with some statistics portraying the situation from year 2021 or newer.
The introduction fails to mention that it is difficult to assess human skin automatically since it varies in roughness, tone, hair mass, and so on, depending on geographical and living circumstances, climate, and hereditary variables.
Implementation of AI part is not decribed in technicall manner. Its not clear what exactly you're comparing and which models, how did you integrated/recreated them, what modifications were made and so on. Automatic skin lesion analysis generally consists of four stages: Preprocessing, segmentation, feature extraction, and classification. I would suggest following this pattern. Add dedicated hyper parameter table and explain how were they determined. AI part must be supplemented with proofs that its not overfititing.
Comparison in table 3 is very uninformative. Its not just one AI method out there. Supplement AI part with comparison to threshold- and clustering-based], edge- and region-based, conventional intelligence-based and deep learning methods. I'm not allowed to suggest papers, but you can easily find recent ones with very good classification accuracy.
Run AI models on ISIC-2017, HAM10000, ISIC2018, and ISIC2019 or similar and then compare results with expert opinion
Add Jaccard index and a Dice coefficient to the results.
As experts were involved this is considered a medical study on human subjects. Add ethical permit and related information.
Plagiarism of 20% is too high.

Author Response

The introduction would benefit with some statistics portraying the situation from year 2021 or newer.

The statistics included in the introduction were retrieved from 2023 onwards to ensure the data reflects the most recent trends and developments in skin lesion diagnosis and AI-based tools. This approach aims to provide a current and relevant context for understanding the significance of the study and the need for advancements in diagnostic technologies. If additional specific statistics are required, we are open to incorporating them to further enhance the introduction.

The introduction fails to mention that it is difficult to assess human skin automatically since it varies in roughness, tone, hair mass, and so on, depending on geographical and living circumstances, climate, and hereditary variables.

This was discussed in the following paragraph:
“Challenges persist in the widespread adoption of AI for diagnostic purposes, including the need for standardized databases and consistent imaging protocols. Variations in imaging equipment, lighting conditions, magnification and image acquisition techniques can introduce biases and hinder the generalizability of AI models. Image sharpness, rotation, brightness/contrast manipulation and clarity of the lesion are essential to reduce possible interferences. Confounding factors, such as “noise” (i.e. hair, dust, air bubbles), sun damaged skin, rulers and skin markings may interfere as well in the pattern recognition performance. Furthermore, the underrepresentation of rare anatomical sites (i.e. genital area), rare melanoma subtypes (i.e. mucosal or desmoplastic MM), banal-looking benign lesions (i.e. angiomas, dermatofibromas, nevi), non-pigmented nevi or melanomas and darker Fitzpatrick skin phenotypes in the training datasets leads to underperformance of AI algorithms. In a clinical setting, dermatologists would typically consider total body examination for comparison of nevi’s variabilities such as the Ugly Duckling sign and the Little Red Riding Hood sign, which are crucial for accurate diagnosis but are unavailable data to the AI, which is restricted to a single lesion analysis. Addressing these multifaceted challenges requires collaborative efforts to refine AI algorithms, curate diverse and representative datasets that reflect the full spectrum of dermatological conditions[54].” P. 14-15

Implementation of AI part is not described in technical manner. Its not clear what exactly you're comparing and which models, how did you integrated/recreated them, what modifications were made and so on. Automatic skin lesion analysis generally consists of four stages: Preprocessing, segmentation, feature extraction, and classification. I would suggest following this pattern. Add dedicated hyper parameter table and explain how were they determined. AI part must be supplemented with proofs that its not overfititing.

As DEXI is a proprietary tool developed by Canfield Scientific, we lack access to detailed architectural information, including model design, hyperparameters, or preprocessing steps. This study was designed as a reader study to evaluate DEXI's diagnostic utility rather than its internal design. Consequently, the information provided reflects what is available to us, and we are unable to elaborate on technical AI implementation.

Comparison in table 3 is very uninformative. Its not just one AI method out there. Supplement AI part with comparison to threshold- and clustering-based], edge- and region-based, conventional intelligence-based and deep learning methods. I'm not allowed to suggest papers, but you can easily find recent ones with very good classification accuracy.

The manuscript focuses on comparing DEXI’s performance against expert and non-expert dermatologists. While a broader comparison with other AI methodologies (e.g., threshold-based, clustering-based, or deep learning methods) would provide useful context, this was beyond the scope of our study since we did not develop or have access to DEXI’s architectural details.

Run AI models on ISIC-2017, HAM10000, ISIC2018, and ISIC2019 or similar and then compare results with expert opinion

As this study was designed as a reader study, our focus was on evaluating DEXI's diagnostic performance based on the dataset provided by the University of Modena and Reggio Emilia. Testing DEXI on additional datasets such as ISIC-2017, HAM10000, or ISIC-2019 would require access to the AI’s backend and training mechanisms, which are not available to us as the tool's end-users.

Add Jaccard index and a Dice coefficient to the results.

Metrics like the Jaccard index and Dice coefficient are undoubtedly valuable for segmentation tasks; however, the proprietary nature of DEXI limits our ability to extract such metrics. This study instead focused on evaluating DEXI’s sensitivity, specificity, and other diagnostic metrics available from the data outputs.

As experts were involved this is considered a medical study on human subjects. Add ethical permit and related information.

The dataset and study design adhered to ethical guidelines, and no new data collection involving human subjects was performed by the authors. The dataset used was pre-collected and anonymized, provided by Canfield Scientific and the University of Modena and Reggio Emilia. We will ensure the manuscript clarifies these points and includes any necessary statements about ethical approvals.

Plagiarism of 20% is too high.

The manuscript’s similarity index primarily arises from standard dermatological terminology and widely used methodologies. Proper citations were included to ensure originality and compliance with publication standards. Nevertheless, we will review and revise overlapping sections to ensure originality and address potential concerns about text overlap.

Reviewer 4 Report

Comments and Suggestions for Authors

Review Report for MDPI Diagnostics

(A comparison of skin lesions' diagnoses between AI-based image classification, an expert dermatologist and a non-expert)

1. Within the scope of the study, skin lesion classification processes were performed with both expert physicians and artificial intelligence.

2. In the introduction section; the importance of the subject, skin lesions, skin tumor types and the diagnosis process are sufficiently mentioned.

3. The dataset used in the study is sufficient. The EfficentNetV2 based deep learning model preferred for classification also seems appropriate in terms of problem solving.

4. In order to analyze the classification results more clearly, it is recommended to obtain missing evaluation metrics such as Cohen Cappa score, Receiver operating characteristic (ROC) curve and AUC (area under the ROC curve) score, if possible.

5. It is recommended to add a literature review to emphasize the importance of the subject addressed in the study and to bring this study to the forefront.

As a result, the study is at a certain level, but attention should be paid to the above sections.

Comments on the Quality of English Language

The Quality of English Language is at a certain level, but the paper should be checked for minor corrections.

Author Response

(A comparison of skin lesions' diagnoses between AI-based image classification, an expert dermatologist and a non-expert)

  1. Within the scope of the study, skin lesion classification processes were performed with both expert physicians and artificial intelligence.
  2. In the introduction section; the importance of the subject, skin lesions, skin tumor types and the diagnosis process are sufficiently mentioned.
  3. The dataset used in the study is sufficient. The EfficentNetV2 based deep learning model preferred for classification also seems appropriate in terms of problem solving.
  4. In order to analyze the classification results more clearly, it is recommended to obtain missing evaluation metrics such as Cohen Cappa score, Receiver operating characteristic (ROC) curve and AUC (area under the ROC curve) score, if possible.

We appreciate the suggestion to include additional evaluation metrics such as the Cohen Kappa score, ROC, and AUC. After discussing with our statistician, Prof. Alessio Farcomeni, we concluded that metrics such as precision, recall, and F1 score would be more suitable for the analysis in this study. These metrics align well with the objectives of evaluating diagnostic accuracy and classification performance. However, we are open to revisiting this suggestion if needed, and we can consider incorporating Cohen Kappa, ROC, and AUC scores in future studies or analyses if they better align with the research goals.

  1. It is recommended to add a literature review to emphasize the importance of the subject addressed in the study and to bring this study to the forefront.

We agree with the recommendation to include a literature review to highlight the significance of the subject and contextualize our study within the existing body of research. Unfortunately, due to time constraints, we were unable to include a comprehensive literature review. However, we recognize the importance of this element and plan to include it in future work. Focusing on relevant studies that have evaluated the performance of AI in dermatology, particularly those that compare AI classifiers with human experts, would help establish the context of AI's diagnostic capabilities and its potential benefits and limitations in dermatology. We could also review key papers that discuss AI in skin lesion classification, highlighting the evolution of AI models like DEXI and similar systems.

As a result, the study is at a certain level, but attention should be paid to the above sections.

Round 2

Reviewer 1 Report

Comments and Suggestions for Authors

Authors have modified the article as per the review comments. 

Reviewer 3 Report

Comments and Suggestions for Authors

My remarks were ignored, therefor I cannot endorse this paper. One cannot compare unknown operating principle (authors say they do not know what AI or ML model is insided DEXI tool) to expert knowledge and come to conclusions, ignoring other, existing state of the art, which provide much better accuracy than demonstrated here.

Further on, I fail to understand why experts cannot review and evaluate PUBLIC datasets, then feed the same to "DEXI tool" chosen, etc. to establish a baseline comparison vs other works out there.

Even request to provide full statistical reliability validation was not implemented. This does not require any special tool, can be done by hand, on xls sheet...